# Research on Residual Stresses and Microstructures of Selective Laser Melted Ti6Al4V Treated by Thermal Vibration Stress Relief

**DOI:** 10.3390/mi14020354

**Published:** 2023-01-31

**Authors:** Shuguang Chen, Jinlong Ma, Hanjun Gao, Yesong Wang, Xun Chen

**Affiliations:** 1School of Mechanical Engineering, Jiangsu University of Science and Technology, Zhenjiang 212100, China; 2State Key Laboratory of Virtual Reality Technology and Systems, School of Mechanical Engineering and Automation, Beihang University, Beijing 100191, China; 3Jingdezhen Research Institute, Beihang University, Jingdezhen 333000, China; 4School of Automation, Jiangsu University of Science and Technology, Zhenjiang 212000, China

**Keywords:** selective laser melting, residual stress, additive manufacturing, thermal-vibration stress relief

## Abstract

The efficient and cost-effective residual stress control method is of great significance for the application of additive manufacturing (AM) technology. In this work, thermal-vibration stress relief (TVSR) with different temperatures and dynamic stresses was performed on Ti6Al4V samples prepared by selective laser melting (SLM), the stress relief effects of TVSR and its influence on phase and microstructure were investigated and compared with thermal stress relief (TSR) and vibration stress relief (VSR), and the stress relief mechanisms of these methods are discussed. It was found that the residual stress relief rate can reach 86.76% after TVSR treatment at a temperature of 380 °C and a dynamic stress of 400 MPa, which increased by 63.63% compared with VSR under the same dynamic stress. The efficiency is increased by 76% compared with TSR at 580 °C and the residual stress relief rate is almost the same. After TVSR, VSR and TSR treatments, the grain morphology, size and phase content of samples were basically unchanged, and low-angle grain boundaries (LAGBs) were increased after TVSR and VSR treatments and decreased after TSR treatment. The results confirm that the TVSR method has the ability to control the residual stress of selective laser melted Ti6Al4V with low time and cost consumption, and are helpful for engineering applications of TVSR.

## 1. Introduction

As one of the materials with high specific strength and excellent mechanical properties, titanium alloy has a wide range of applications in the fields of aerospace, medicine and energy. It is also a typical hard-to-machine material due to its low thermal conductivity and small elastic modulus, which make it easy to produce a rebound phenomenon and aggravate tool wear during machining [1]. Additive manufacturing (AM) is a near net forming technology, which provides a new solution for the manufacture of hard-to-machine metals and alloys like titanium alloy. However, there are still some challenges in the quality control of AM [2]. Uneven and excessive residual stress is one of the prominent and noteworthy problems. During the AM process, the material is in an extreme imbalance of thermal metallurgy similar to welding [3]. This extreme imbalance will introduce residual stress into the part with an amplitude close to the yield strength of the material [4]. Uneven and excessive residual stresses may lead to cracking [5], excessive deformation or even separation of the part from the substrate [6]. In addition, residual stress also has significant effects on the strength, corrosion resistance, crack initiation and propagation, and dimensional stability of materials [7]. Therefore, the control of residual stress is very critical and necessary for AM.

Residual stress can be controlled by optimizing the process parameters and post-stress relief treatment. Optimized process parameters can be obtained through parameter dependence research. Wang et al. [8] found that residual stresses can be effectively reduced with decreased laser power and scanning speed. Mugwagwa et al. [9] characterized residual stresses by deformation of a cantilever and found that the increase in laser power and scanning speed led to an increase in deformation of the cantilever. Lai et al. [10] and Liu et al. [11] reported that the lower the scanning speed, the greater the residual stress. The research results of Ali et al. [12] show that layer thickness is also an important factor affecting the residual stress. Levkulich et al. [13] found that the residual stress on the top surface of the sample decreases with an increase in the height of the part. Casavola et al. [14] reported that residual stresses in AISI 18 Maraging steel first increase and then decrease with sample height. Rani et al. [15] found that the distribution of residual stresses is related to the path direction, and longitudinal stresses are more dominant than transverse stresses. Chen et al. [16] found that solid phase transformation is also one of the factors that cannot be ignored for the formation of residual stress in the AM process of Ti-6Al-4V. Residual stresses can be reduced by optimizing the process parameters, but inevitably exist in AM samples as a result of the combined action of the temperature gradient mechanism, solid-state phase transition mechanism and cool-down phase mechanism [2]. Therefore, stress relief treatment of AM samples is an indispensable process.

At present, the most commonly used stress relief method for AM samples is thermal stress relief (TSR). Li et al. [17] conducted TSR treatment on AM 316 L stainless steel at 400 °C for 2 h and found that the average residual stress decreased by 53.7%. Song et al. [18] reported that the residual stress of iron was almost completely eliminated with TSR treatment at 640 °C for 2.5 h. Leuders et al. [19] found a very high residual stress gradient with highest stresses in the as-built Ti-6Al-4V sample, and the residual stresses were reduced to about 10 MPa after TSR treatment at 800 °C for 2 h. It should be mentioned that titanium alloys are easily oxidized at high temperatures [20]. Consequently, TSR treatments of titanium alloys are usually carried out in a vacuum furnace or an atmosphere protection furnace, which greatly increases the cost of residual stress relief treatment. In addition, TSR treatments are time-consuming due to long-period heating and cooling. Vibration stress relief (VSR), which utilizes the principle that dynamic stress and local residual stress superimpose to produce plastic deformation in higher stress areas, has the ability to rapidly eliminate residual stress in materials. Sun et al. [21] performed VSR treatment on 35# bar steel and found that residual stress decreased by about 48%. Gao et al. conducted VSR treatment on 7075-T651 aluminum alloy [22] and Ti-6Al-4V [23]. The results show that residual stresses can be reduced by VSR, but the effect depends on the magnitude of the dynamic force.

Recently, a thermal-vibration stress relief (TVSR) method was proposed and has aroused extensive interest from researchers [24,25,26,27]. Song et al. [25] reported that the circumferential and axial residual stresses of large aluminum alloy rings were released by 44.43% and 45.14%, respectively. Wu et al. [26] found that the maximum reduction of stress in SiCP/Al reduced by 81%. Gao et al. [27] studied the stress relief effect of TVSR on titanium alloy; the results illustrated that the stress relief rate of TVSR with a 300 MPa dynamic stress at 280 °C can reach more than 90%, which is 45.04% higher than VSR with the same dynamic stress, and the efficiency of TVSR is 4.2 times that of TSR with a similar stress relief effect.

As one of the key AM post-treatment processes, the reduction in time and cost consumption of stress relief treatment is of great significance for controlling the manufacturing cost of AM titanium alloys and promoting their further application. It can be seen from the above research that among the commonly used stress relief methods, TVSR shows great advantages in efficiency and effect. However, whether this method is effective for AM titanium alloy is still unclear. It should be noted that the oxidation of titanium alloy is not significant below 550 °C [28]. This shows that vacuum or inert gas protection is unnecessary for TVSR, and this can effectively reduce the cost of residual stress relief. In order to fill in this research gap and determine the suitability of TVSR for AM titanium alloys under non-vacuum or non-inert-gas protection environments, in this study, selective laser melting (SLM) was used to prepare Ti6Al4V specimens, TVSR treatments with different temperatures and dynamic stresses were conducted on the samples to assess the stress relief effect, and the results were compared with TSR and VSR. The influences of those stress relief methods on phase and microstructure are investigated, and the stress relief mechanisms are also discussed.

## 2. Experimental Methods

### 2.1. Specimen Preparation

In this study, the Ti-6Al-4V material was fabricated using a BLT-A300 (BLT, Xi’an, China) system equipped with a Yb-fiber laser with a 500 W nominal maximum power with particle size in the range of 18–65 µm. All samples were first produced with a cylindrical geometry with a diameter of 16 mm and a height of 91.6 mm directly on a substrate (Figure 1a). Laser power 250 W, scanning speed 1250 mm/s, single-layer powder thickness 60 μm and preheated temperature 80 °C were adopted when the samples were produced. The scanning strategy was unidirectional scanning within a single layer, and with a rotation of 66.7 °C in next layer, which is also shown in Figure 1a. The entire manufacturing process was carried out in an inert gas-argon environment.

Selective laser melted (SLMed) Ti-6Al-4V cylinders BLT-A300 (BLT, Xi’an, China) were then cooled in the furnace after 825 °C/2 h vacuum heat treatment and separated from the substrate by wire cutting, and finally machined into threaded cylindrical test specimens as shown in Figure 1b with a cemented carbide tool. The spindle speed was 1000 r/min, the cutting depth was 1 mm and the feed speed was 0.5 mm/min.

### 2.2. Residual Stress Measuring Setup

There are many residual stress testing methods with clear measurement theory and verified by a large number of experiments [2,29]. According to the shape of the sample, the residual stresses of the SLMed Ti6Al4V sample were measured by an X-ray diffraction stress meter (Proto, Ontario, Canada), which calibrates with Cu target. The parameters used in the measurement are as follows: spot diameter = 2 mm, exposure time = 2.5 s, distance from X-ray detector to measuring point = 200 mm, the swing angle of the detector = ±25 °C and the sampling interval = 6.25 °C. As shown in Figure 2, the residual stress measuring point was located at the middle of the specimens, and the axial and circumferential residual stresses were measured before and after stress relief treatments.

### 2.3. Specimen Processing

In order to quantitatively control the vibration and temperature parameters, all stress relief experiments were performed on the HYG-300 high-temperature and high-frequency fatigue testing machine produced by Changchun Haoyuan Testing Machine Co., Ltd., Changchun, China. As shown in Figure 3, HYG 300 high-temperature and high-frequency fatigue testing machine includes vibration module and thermal module. In the vibration module, the weight of the upper fixture and counterweight is 480 kg, the maximum dynamic load that can be provided is 300 kN, the vibration frequency range is 50~350 Hz, the frequency resolution is 0.1 Hz, the static force accuracy is ±0.5% and the dynamic force accuracy is ±2%. The heat module is composed of heating wire, asbestos insulation layer and metal shell, which can provide a maximum temperature of 800 °C and a temperature control accuracy of ±1 °C.

In order to evaluate the effect of TVSR and compare with TSR and VSR, as shown in Figure 4, 6 groups of samples were prepared for stress relief treatment with reference to the process parameters used by Gao et al. [27]. In fact, TSR and VSR can be regarded as special TVSR treatments with dynamic stress of 0 MPa and room temperature, respectively. Specimen #1 represents VSR treatment with dynamic stress of 400 MPa, #2 is arranged to perform TSR treatment at 580 °C and #3~#5 represent TVSR under different dynamic stresses and temperatures. The effects of heat and vibration were studied by comparing #1, #4 and #6.

### 2.4. Material Characterization Techniques

Circular table samples 6 mm high were taken from the untreated specimen, specimen #1 (VSR: 400 MPa), specimen #2 (TSR: 580 °C) and specimen #5 (TVSR: 380 °C, 400 MPa), and cut into two parts along the central axis for XRD phase analysis and EBSD observation. The samples used for XRD observation were ground with 400, 800 and 1500 mesh sandpaper and analyzed with an EMPYREAN-2 (PANalytical, Almelo, Netherlands) X-ray diffractometer. The acceleration voltage used in the experiment was 40 kV, the scanning mode was step-by-step, the scanning step was 0.01 °C, the scanning rate was 1.5 °/min, and the scanning angle was in a range of 30~80 °C.

JSM-F100 field-emission scanning electron microscope (JEOL Ltd., Tokyo, Japan) was used to observe the grain and phase. The EBSD test has high requirements for the surface quality of the samples; therefore, electrolytic polishing was carried out with a voltage of 30 V. The polishing solution was a mixture of 95% glacial acetic acid and 5% perchloric acid. The polishing time was about 25 s. The acceleration voltage used during scanning was 20 kV, and the scanning step was 0.1 μm, the magnification was 2000×, and the data were processed by OIM7.0 software.

## 3. Results and Discussion

### 3.1. Residual Stress

The axial and circumferential residual stresses of SLMed Ti-6Al-4V samples before and after TSR, VSR and TVSR treatments are shown in Figure 5, and the values are also given in Table 1. The axial and circumferential residual stresses of untreated specimens are all compressive. The axial residual stresses are in a range of −415.09~−489.11 MPa with an average value of −465.94 MPa and a standard deviation of 21.18 MPa. The circumferential residual stresses range from −244.27 to −332.24 MPa with an average stress equal to −286.57 MPa and a standard deviation of 34.96 MPa. This indicates that the initial stresses of the specimens were uniform, and the value of the axial residual compressive stress was about twice that of the circumferential direction.

After VSR treatment (#1) with a dynamic stress of 400 MPa, the amplitude of axial residual stress of the SLMed titanium alloy Ti-6Al-4V decreased from 415.09 to 319.06 MPa with a relief rate of 23.13%. The magnitude of circumferential residual stress decreased from 315 to 284 MPa with a relief rate of 10.02%. For the samples treated by TSR (#2), the magnitude of axial residual stress decreased from 488.26 to 16.38 MPa with an elimination rate of 96.65%. The magnitude of circumferential residual stress decreased from 251.3 to 27.63 MPa, and the residual stress relief rate was 89.01%.

When 300 MPa dynamic stress was used for TVSR treatment at 280 °C (#3), the axial residual stress amplitude of the sample decreased from 483.96 to 179.62 MPa, with a relief rate of 62.89%. The magnitude of circumferential residual stress decreased from 244.27 to 125 MPa with a relief rate of 48.83%. When the dynamic stress was kept unchanged and the temperature increased to 380 °C, the magnitude of axial residual stress decreased from 460.11 to 332.24 MPa, with a relief rate of 72.91%. The magnitude of circumferential residual stress decreased from 332.24 to 72.31 MPa with an elimination rate of 78.24%. When the temperature was maintained at 380 °C and the dynamic stress was increased to 400 MPa, the magnitude of axial residual stress decreased from 489.11 to 64.78 MPa, with an elimination rate of 86.76%. The magnitude of circumferential residual stress decreased from 259.25 to 64.13 MPa, with a relief rate of 75.26%. When the dynamic stress was 0 MPa at 380 °C, the residual stress could also be eliminated. The magnitude of axial residual stress decreased from 459.1 to 277.05 MPa, with a relief rate of 39.46%. The magnitude of circumferential residual stress decreased from 316.72 to 186.55 MPa, with a relief rate of 41.10%.

These results indicate that all of the VSR, TSR and TVSR treatments can eliminate the residual stress of SLMed Ti-6Al-4V samples. The axial residual stress in the sample was larger and the vibration form of VSR and TVSR treatment is axial vibration; therefore, the axial residual stress is discussed here. VSR treatment with a dynamic stress of 400 MPa shows that the axial residual stress relief rate is only 23.13%, indicating that it is difficult for VSR to eliminate the residual stress of SLMed Ti-6Al-4V, while the axial residual stress relief rate of TSR treatment can reach 96.65%, but the whole treatment period reaches 25 h. At 380 °C, the axial residual stress relief rate of TVSR treatment with dynamic stress of 400 MPa reaches 86.76%, with residual stress elimination which is similar to TSR, but the time spent is only 6 h and the efficiency is increased by 76%. However, at the same temperature, without dynamic stress, the axial residual stress relief rate is only 39.46%, which indicates that TVSR treatment is better than TSR, VSR or even their superposition with the same parameters.

Compared with the results of Gao et al. [27], it can be found that under the same process parameters, the residual stresses of SLMed Ti-6Al-4V are more difficult to control than those of extruded Ti-6Al-4V. With the same VSR treatment with a dynamic stress of 400 MPa, the axial residual stress relief rate of extruded Ti-6Al-4V can reach 48.76%, while the residual stress relief rate of the SLMed Ti-6Al-4V is only 23.12%. TVSR treatment also has a similar phenomenon. When TVSR treatment is carried out at 280 °C and 300 MPa, the axial residual stress relief rate of extruded Ti-6Al-4V reaches 93.80%, while that of the SLMed sample is only 62.89%. After heat treatment at 580 °C/8 h, the axial residual stress change rates of extruded and SLMed Ti-6Al-4V are 112.47 and 103.35%, respectively.

### 3.2. Phase Analysis

Qualitative analyses of the phase transition within the samples under stress relief treatments were performed with recourse to the XRD experiments, and quantitative analyses were performed with EBSD experiments. The X-ray diffractograms depicted in Figure 6 clearly show that there is only a small amount of β phase in all samples; α phase is the dominant fraction. It should be mentioned that α and α′ phases have similar crystal structures and lattice parameters. Therefore, they cannot be distinguished from the XRD spectra [30]. However, combined with the observation results of the microstructure, it can be determined that the dominant phase in the samples is α phase; the specific judgment basis is discussed in the next section.

Figure 7 shows the EBSD phase maps. The α phase and β phase are colored green and red, respectively. The results are consistent with the XRD experiments. 

The content of α phase in the sample without stress relief treatment (Figure 7a) is 99.7%; after TVSR treatment (Figure 7b), the content of α phase is 99.5%. In the samples of VSR (Figure 7c) and TSR (Figure 7d) treatments, the content of α phase is 99.2 and 99.6%, respectively. The maximum difference of phase content of the four samples is only 0.5%, which indicates that stress relief treatment on SLMed Ti-6Al-4V samples basically does not change the phase content and distribution.

### 3.3. Microstructure

To investigate if preferential orientation of the α platelet developed before and after stress relief treatments, inverse pole figure (IPF) maps performed by EBSD are presented in Figure 8. Visual analysis of the IPF maps suggests that α platelets are randomly oriented throughout the sample and with similar orientations in local. Unlike the results of Beladi et al. [31], no acicular α′ martensite is observed within the untreated sample (Figure 8a). This phenomenon is due to all specimens being heat-treated at 850 °C/2 h before stress relief treatment in order to promote the full diffusion of the aggregated elements caused by the rapid cooling rate in the process of SLM; acicular α′ martensite transitions to α platelets during this process. Vrancken et al. [32] conducted the same heat treatment as this study for SLMed Ti-6Al-4V, and the microstructure is also similar. However, no columnar crystals are observed in Figure 8a, which may be due to the higher magnification used in EBSD observation in this study. Furthermore, Figure 8a provides evidence for the judgment that α phase is the dominant fraction in Section 3.2. Figure 8b,d illustrate that there are no obvious differences in grain shape before and after stress relief treatment.

Figure 9 shows the EBSD analysis results of the grain sizes of specimens #0 (untreated), #1 (VSR: 400 MPa), #2 (TSR-580 °C) and #5 (TVSR-380 °C–400 MPa). The equivalent grain diameters of all samples are mostly less than 10 μm, and the maximum difference between the average grain area of each sample is only 0.85 μm^2^, but the length of α platelets is mostly in the range of 10~20 μm. Therefore, the width of α platelets hardly changed after stress relief treatments.

Figure 10 presents low-angle grain boundary (LAGB, 2 to 15 °C) distribution, high-angle grain boundary (HAGB, 15 to 180 °C) distribution and the misorientation angle before and after stress relief treatments. HAGBs are located at the edges of grains, and LAGBs are distributed in grains. As is shown in Figure 10, the LAGB of the untreated sample was 31.5%. The LAGB is composed of dislocations, which are formed mainly during forming and machining. SLMed Ti-6Al-4V without heat treatment is mainly composed of martensite α′. A large number of dislocations appear in martensite during the phase transition of β to α′ [33,34]. After heat treatment at 800 °C/6 h for SLMed Ti-6Al-4V, Chao et al. [35] still observed high-density dislocations in the grains. It can be seen from Figure 10b that after TVSR treatment, the LAGB in some grains increased significantly, accounting for 49.6% in total, and increased by 18.1% compared with the sample without any treatment. Figure 10c shows that VSR treatment also led to a slight increase in LAGB. After VSR treatment, the proportion of LAGB was 34.5%. However, in the sample treated with TSR (Figure 10d), the LAGB decreased significantly, with a fraction of only 26%. It can be seen from Figure 10e that the misorientation angles of all samples have similar laws in distribution. The misorientation angles are concentrated at 60, 68 and 86 °C, and this is not consistent with the distribution law and theoretical proportion of general titanium alloy. This indicates that the preferred orientation of crystals appears in the samples. The phenomenon of the preferred orientation of crystals may be as a result of variant selection in the process of transition of β to α′ or α under defects, residual stress, dislocation and β column crystal [36].

LAGBs are mainly composed of dislocations, so LAGB changes are mainly caused by the movement, multiplication or annihilation of dislocations. Dislocation is an important carrier of plastic deformation, and its movement and evolution are signs of plastic deformation of material. The phenomenon that LAGB changes in Figure 10 concentrate on a part of the grains illustrates that local plastic deformation occurs during stress relief treatment.

When TSR treatment is conducted, the thermal motion of atoms becomes more intense because of the increase in temperature, and the possibility of atoms with the wrong arrangement returning to the equilibrium position is increased. When the temperature is higher than a certain range, the dislocations formed due to the wrong arrangement of atoms will be significantly reduced by annihilation. In the process of VSR treatment, when the vector sum of dynamic stress and residual stress is greater than the minimum stress for dislocation actuation, the movable dislocation starts to move, and the pinned dislocation becomes the F-R source and leads to dislocation multiplication. During TVSR treatment, there is a competition mechanism between dislocation annihilation and multiplication in the material.

Theoretically, the dislocation density of samples treated by TVSR should be between those of TSR and VSR. Because the temperature of TVSR treatment of SLMed Ti-6Al-4V in this study is lower than the temperature of TSR, although the thermal movement of metal atoms can be enhanced, the threshold value of the energy required for the wrong arrangement of atoms to return to the equilibrium position has not been reached, and the immovable dislocations are still in the pinning state. Therefore, the same phenomenon of dislocation multiplication as that of VSR can be observed in the sample treated with TVSR. However, under the same dynamic stress, due to the larger atomic spacing and lower yield strength, local plastic deformation may occur more easily, and the dislocation density is correspondingly larger than that of VSR.

### 3.4. Stress Relief Mechanism

During TSR, VSR and TVSR treatments, if creep, solid phase transformation or plastic deformation caused by other reasons occur under the action of thermal movement or stress, resulting in a reduction in elastic strain, the residual stress will be reduced. In essence, TSR can be regarded as the stress relaxation behavior of materials at a certain temperature. Cottam et al. [37] conducted heat treatment of laser-clad Ti-6Al-4V at 450 °C, and the microstructure analysis showed that phase transition between α and β leads to the relaxation of residual stress. In this study, the phase content did not change before and after TSR treatment. Elmer et al. [38] also found that Ti-6Al-4V exhibited stress relaxation at 550 °C without phase transition. When no solid phase transformation occurs, the reduction of residual stress is generally caused by creep.

In atomic scope, with the increase in temperature, the thermal movement of atoms increases, vacancy defects gradually diffuse from the tensile stress zone to the compressive stress zone and the strain energy of grains is released. This phenomenon is called the grain boundary diffusion mechanism and intragranular diffusion mechanism. These diffusion mechanisms are microscopically manifested as the rotation of grains and the sliding of grain boundaries, respectively [39]. The microstructures of Ti-6Al-4V samples in this study are mainly composed of a lath; therefore, grain rotation becomes particularly difficult. The misorientation angles of all samples have similar laws in distribution shown in Figure 10e, which also proves this inference. Therefore, the decrease in residual stresses caused by TSR treatment in this study is mainly caused by grain slip on the microscale and by the atomic diffusion mechanism on the atomic scale.

The dislocation density in local grains increased after VSR treatment, indicating that microplastic deformation occurred. Metals are composed of many grains with different orientations. Due to the orientation and stress state of each grain being different, the dislocation movements of the grains are also different, and dislocation slip and multiplication first occur in grains with large residual stress and favorable orientation. With the slip of dislocation, the plastic deformation of those grains increases and the residual stress in grains decreases, and the rotation and sliding of grains will appear inevitably in the surrounding grains to ensure the continuity of material. Parts of grains with unfavorable orientations may turn to favorable orientations, resulting in micro-yielding.

Theoretically, the TVSR mechanism should be the unification and competition of the TSR and VSR mechanisms. Which of these is the main mechanism depends on the process parameters. In this study, the temperature adopted in TVSR treatment was lower than the recommended temperature range of TSR; thus, residual stress decrease was mainly caused by the dislocation slip mechanism as in VSR. In comparison to VSR, due to the increase in temperature, the thermal motion of atoms is enhanced, the critical resistance of dislocations decreases in TVSR treatment, and dislocation slip and multiplication are easier. In addition, the cross-slip ability of screw dislocations increases with temperature, and the edge dislocations can also climb to overcome obstacles. The abilities of grain rotation and sliding are also enhanced compared with VSR, and the probability of grains changing from unfavorable orientation to favorable orientation is also increased. Therefore, under the same dynamic stress, TVSR treatment has a higher dislocation density and a better residual stress elimination effect compared with VSR.

## 4. Summary and Conclusions

In this paper, the TVSR method and mechanism of SLMed Ti-6Al-4V are studied and compared with TSR and VSR in the residual stress relief effect. XRD and EBSD characterizations were carried out to investigate these stress relief methods on phase and microstructure. Finally, the stress relief mechanisms are also discussed. The primary contributions and conclusions of this work are as follows:After TVSR treatment at a temperature of 380 °C and a dynamic stress of 400 MPa, the amplitude of residual stress is reduced from 489.11 to 64.78 MPa and the residual stress relief rate can reach 86.76%, which is increased by 63.63% compared with VSR under the same dynamic stress. The residual stress relief efficiency is increased by 76% compared with TSR at 580 °C and the residual stress relief rate is almost the same.After TVSR, VSR and TSR treatments, the grain morphology, size and phase content of SLMed Ti-6Al-4V are basically unchanged.After TVSR, VSR and TSR treatments, the LAGB content of SLMed Ti-6Al-4V changed significantly. After TVSR and VSR treatments, the LAGBs increased by 18.1 and 3.0%, respectively, and were concentrated in the local grain range. The LAGBs decreased by 5.5% after TSR treatment.

## Figures and Tables

**Figure 1 micromachines-14-00354-f001:**
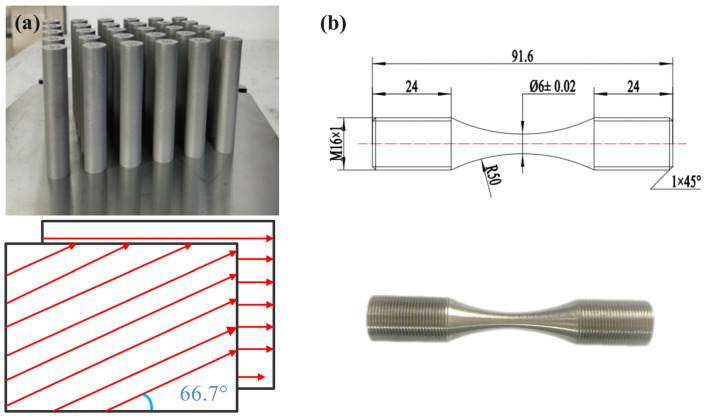
(**a**) The fabricated cylinders Ti-6Al-4V and scanning strategy; (**b**) the threaded cylindrical test specimens.

**Figure 2 micromachines-14-00354-f002:**
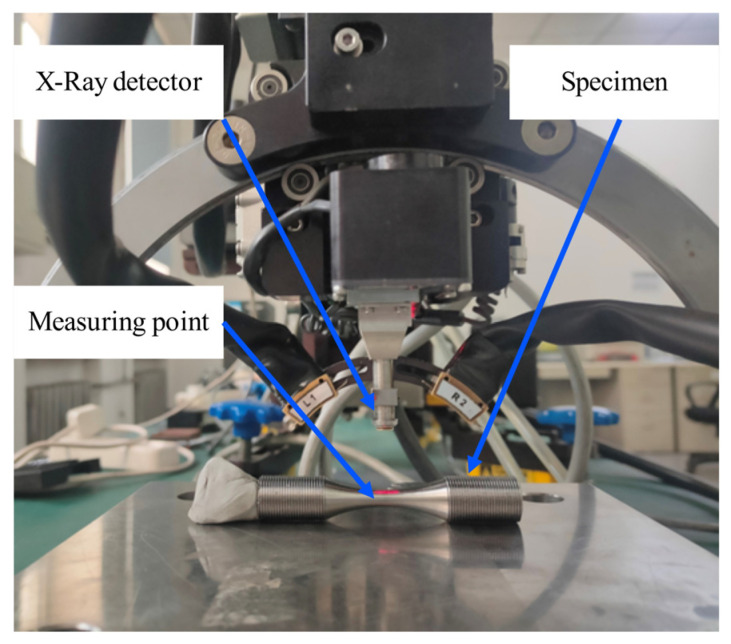
X-ray diffraction stress meter and residual stress measuring point.

**Figure 3 micromachines-14-00354-f003:**
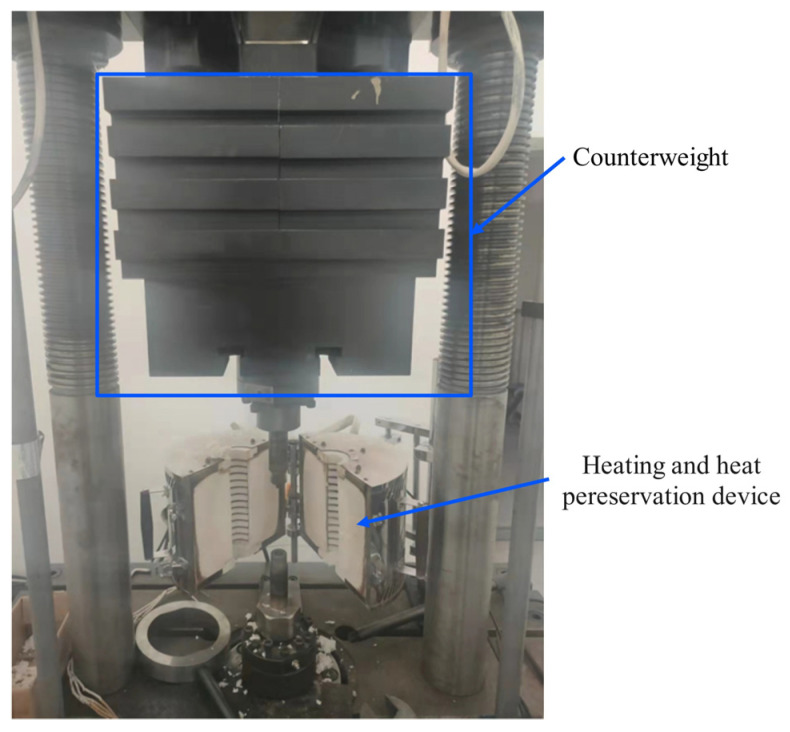
High-temperature and high-frequency fatigue testing machine for residual stress relief experiments.

**Figure 4 micromachines-14-00354-f004:**
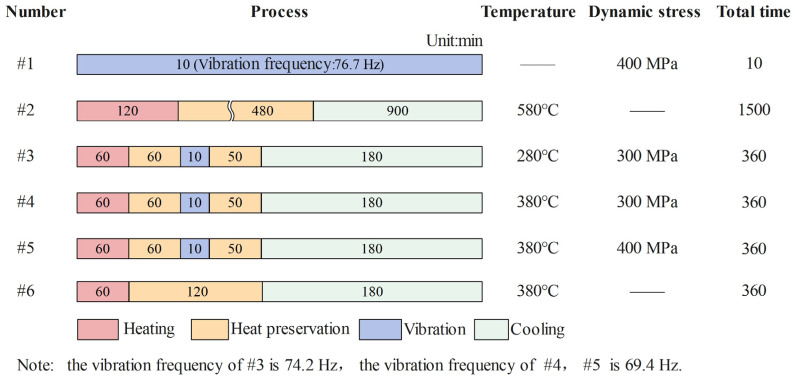
Arrangement of residual stress relief experiments.

**Figure 5 micromachines-14-00354-f005:**
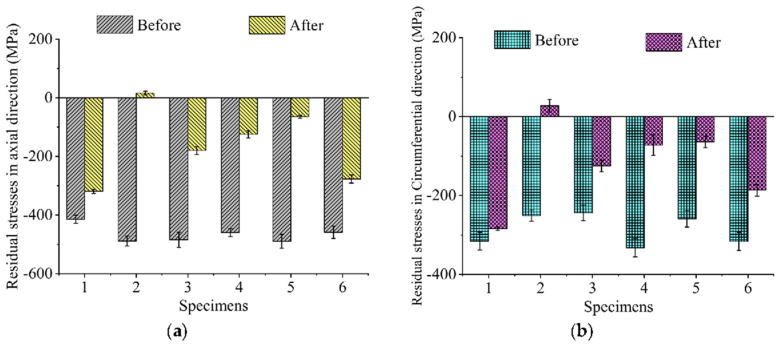
Residual stress measurement results before and after the stress relief treatments in (**a**) axial direction and (**b**) circumferential direction.

**Figure 6 micromachines-14-00354-f006:**
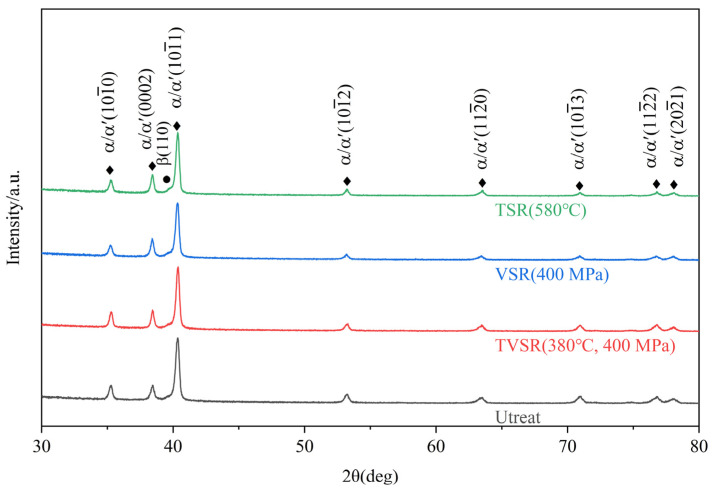
XRD patterns of SLMed Ti6Al4V samples before and after stress relief treatments.

**Figure 7 micromachines-14-00354-f007:**
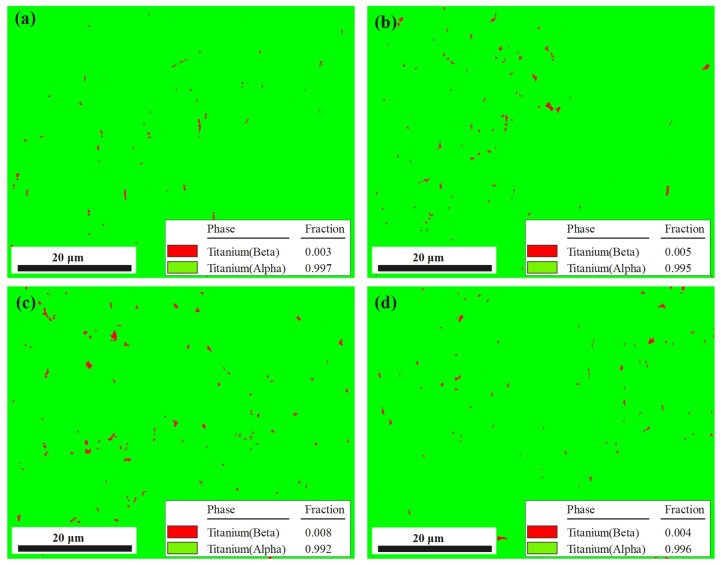
The phase distributions and contents in samples before and after stress relief treatments: (**a**) #0 (untreated); (**b**) #5 (TVSR: 380 °C, 400 MPa); (**c**) #1 (VSR: 400 MPa); (**d**) #2 (TSR: 580 °C).

**Figure 8 micromachines-14-00354-f008:**
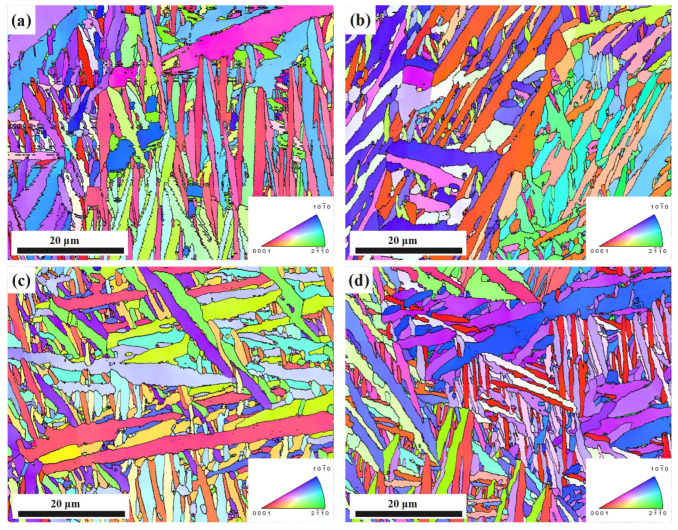
The IPF maps before and after stress relief treatments: (**a**) #0 (untreated); (**b**) #5 (TVSR: 380 °C, 400 MPa); (**c**) #1 (VSR: 400 MPa); (**d**) #2 (TSR: 580 °C).

**Figure 9 micromachines-14-00354-f009:**
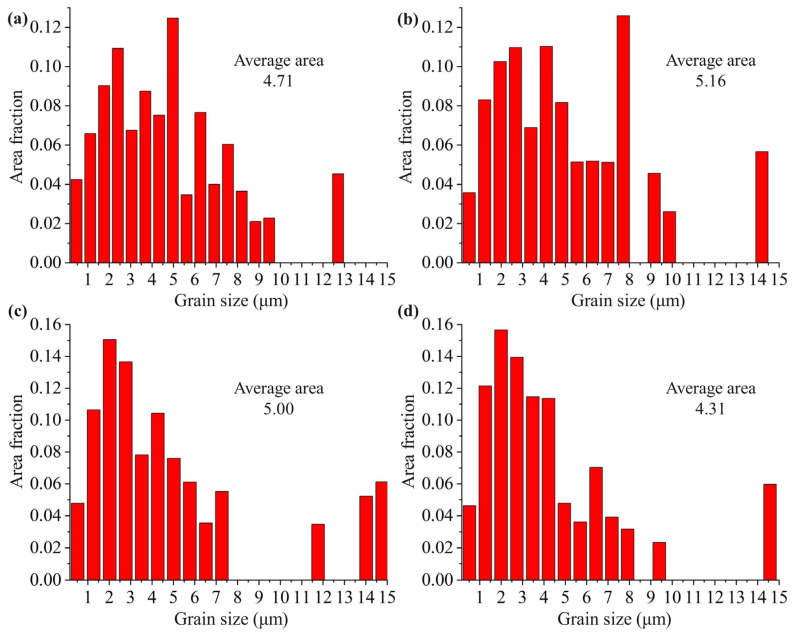
EBSD analysis of grain size: (**a**) #0 (untreated); (**b**) #5 (TVSR: 380 °C, 400 MPa); (**c**) #1 (VSR: 400 MPa); (**d**) #2 (TSR: 580 °C).

**Figure 10 micromachines-14-00354-f010:**
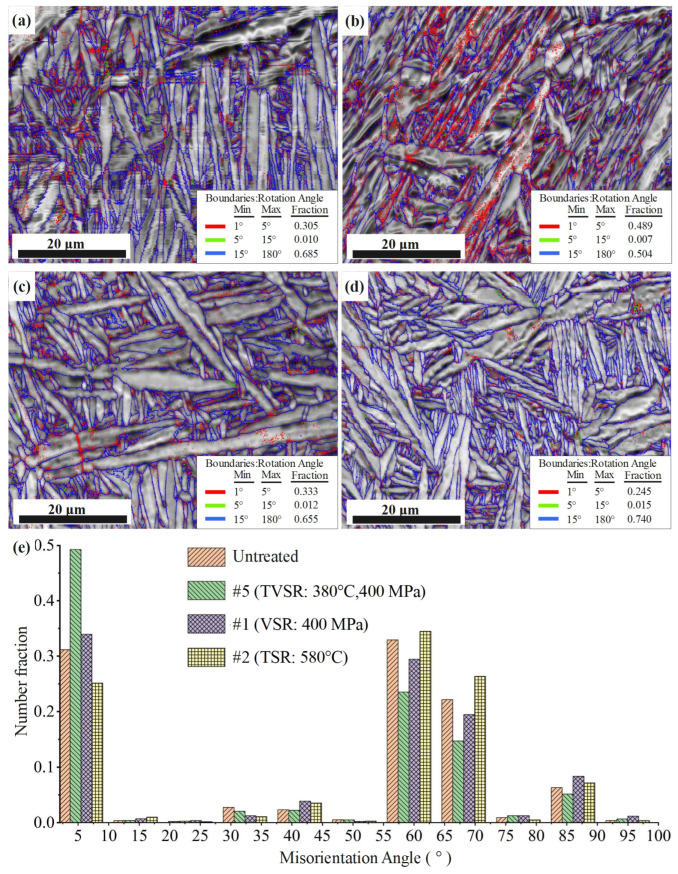
EBSD analysis of rotation angle: (**a**) #0 (untreated); (**b**) #5 (TVSR: 380 °C, 400 MPa); (**c**) #1 (VSR: 400 MPa); (**d**) #2 (TSR: 580 °C); (**e**) misorientation angle.

**Table 1 micromachines-14-00354-t001:** The values of residual stress measured by X-Ray Diffraction Stress Meter.

Num.	Residual Stresses in Axial Direction/MPa			Residual Stresses in Circumferential Direction/MPa		
	Before	After	Stress Relief Rate	Before	After	Stress Relief Rate
#1	−415.09 ± 13.98	−319.06 ± 7.26	23.13%	−315.61 ± 22.14	−284 ± 4.57	10.02%
#2	−488.26 ± 16.62	16.38 ± 5.93	96.65%	−251.3 ± 13.81	27.63 ± 15.69	89.01%
#3	−483.96 ± 25.63	−179.62 ± 13.7	62.89%	−244.27 ± 20.11	−125 ± 14.68	48.83%
#4	−460.11 ± 12.56	−124.64 ± 12.88	72.91%	−332.24 ± 22.97	−72.31 ± 26.28	78.24%
#5	−489.11 ± 23.81	−64.78 ± 5.28	86.76%	−259.25 ± 21.14	−64.13 ± 14.49	75.26%
#6	−459.1 ± 20.58	−277.05 ± 13.84	39.46%	−316.72 ± 22.76	−186.55 ± 14.86	41.10%

## Data Availability

The data that support the findings of this study are available from the corresponding author upon reasonable request.

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
