# Peer review of "Research on Residual Stresses and Microstructures of Selective Laser Melted Ti6Al4V Treated by Thermal Vibration Stress Relief"

_micromachines, 2023, doi:10.3390/mi14020354_

Round 1

Reviewer 1 Report

It seems that the manuscript contains a serious methodological error. In Experimental section, it is stated “Then, selective laser melted (SLMed) Ti-6Al-4V cylinders are cooled in the furnace after 825 ℃/2h vacuum heat treatment and separated from the substrate by wire cutting, and finally machined into threaded cylindrical test specimens as shown in Fig. 1(b) with a cemented carbide tool, the spindle speed is 1000 r/min, the cutting depth is 1 mm, and the feed speed is 0.5 mm/min”. Thus, the residual stresses that arose during the manufacture of the samples by the SLM method were significantly changed. A natural question arises, what, in fact, has been investigated?

Reviewer 2 Report

1.       Shorten the length of the abstract section and add only key information in abstract section.

2.       Discuss the Novelty and clear application of the work in abstract as well as in introduction section.

3.       Shorten the length of the introduction section and add key published work and try to make a bridge between current and previous published work: https://doi.org/10.3390/ma15207094; https://doi.org/10.1016/j.matlet.2020.128347; https://doi.org/10.1016/j.acme.2018.02.007.

4.       Discuss about various methods used for residual stresses evaluation.

5.       Also add the image of the experimental setup along with complete procedure.

6.       Upto what depth someone can use XRD to measure the residual stresses?

7.       Microstructure characterization presented well with good depth discussion. But relation between microstructure and residual stresses need a technical discussion.

8. What standard was used to prepare the specimen?

9. Please mention about the selection of the heat treatment.

10. Try to compare the results with other published work using other methods.

11. In sample 2, residual stress nature was tensile after the heat treatment, why?

12. What are the factors responsible for stress relief?

13. If possible add quantitative measurement of phases in XRD analysis.

14. References are upto the mark but add some references in discussion section to compare the results.

15. Experimental section is presented well.

16. In conclusion section add quantitative results also.

Round 2

Reviewer 1 Report

Dear Authors. The question have been answered properly, and I recommend this manuscript for publication.